# Stress among Students and Difficulty with Time Management: A Study at the University of Galați in Romania

Daniel Lovin [1] and Denis Bernardeau-Moreau [2,*]

[1] STAPS Department UFR STAPS, University of Rennes 2, 35044 Rennes, France
[2] Multidisciplinary Research Unit for Sport, Health, Society (URePSSS), University of Lille, ULR 7369, 59000 Lille, France
* Correspondence: denis.bernardeau-moreau@univ-lille.fr

**Abstract:** Stress is a defining trait of our modern societies. The correlations between economic and social developments and the state of ill-being of populations have long been demonstrated. Today, negative environmental factors such as climate change, war and health crises have consequences on populations. Regardless of gender or age, more and more people are suffering from stress, of which there are many effects. According to studies, stress is an emotional response resulting from human–environmental interaction. They define stress as a state of discomfort and tension caused by external factors. This author believes that stress has a negative impact, which leads to frustration and increased difficulty in addressing issues. Authors describes stress as a physical and psychological state experienced by someone facing a real or potential. For students, stress can be caused by a number of factors. Some of these may include how the university functions, exam periods, relationships with teachers, the pressure that parents put on academic achievement, competition with other students, financial problems, uncertainty about the future, and lack of self-confidence. Symptoms of stress are now well identified. Students suffering from stress report feeling unhappy, having stomach pains, difficulty relaxing and sleeping, mental health problems, and even depression. When it comes to stress, Romanian students are no exception, especially because of the particularly volatile situation faced by the nation. The country has high immigration of skilled and qualified labour, inflation, and depreciation of the national currency. Students also claim to be stressed by the challenges of student life and the demands of the working world. They have a constant fear of failure and doubt themselves, their academic skills, and success in their careers after graduating. Our sociological study aims to deepen our knowledge in this field in Romania. In an already anxious context, it successively examines stress factors, symptoms experienced by students, conditions for organising studies, and ways to improve students' quality of life. To conduct our study, we sampled students at the Dunărea de Jos University of Galați. We analysed 151 questionnaires sent to a sample of students listed in various first-year university courses. The results of this survey ultimately allow us to better identify the time-related, financial, and social factors of stress and the resulting symptoms. It is very clear that the COVID-19 pandemic has had a detrimental effect on this already fragile young population. Our study is also an opportunity to discuss ways to better manage student stress.

**Keywords:** student stress; symptoms of stress; exam periods; uncertainty about the future; lack of self-confidence; fear of failure; detrimental effect of pandemic

## 1. Stress among Students: A Societal Phenomenon

Stress is a defining trait of our modern societies (Dill and Henley 1998; Benton et al. 2003; Dusselier et al. 2005; Kinman and Jones 2005; Owen-Yeats 2005; Robotham and Julian 2006; Roberti et al. 2006; Laurence et al. 2009; Hystad et al. 2009; Collins et al. 2010; Campagna 2013; Riley and Park 2013; Lungu 2021). Regardless of gender or age, more and more people are suffering from stress (Pearlin 1989; Pearlin and Johnson 1977; Menaghan 1982), of which there are many causes and effects. According to Bull (1991),

stress is an emotional response resulting from human–environmental interaction. Golu (1981) defines stress as a state of discomfort and tension caused by external factors. This author believes that stress has a negative impact, which leads to frustration and increased difficulty in addressing issues. Several studies have analysed the psychosocial factors associated with resilience to stress (Garmezy et al. 1984; Werner and Smith 1992; Masten and Coatsworth 1998; Luthar and Cicchetti 2000; Southwick et al. 2005). They have identified several psychosocial resilience factors: positive emotions (optimism, humour); cognitive flexibility (acceptance); search of meaning (through religion, spirituality); and physical activity (through exercise, sport and training). Sillamy and Gavriliu (2009) describe stress as a physical and psychological state experienced by someone facing a real or potential threat. Recent international studies (Beck et al. 2007; Conley et al. 2013; Eicher et al. 2014; Buskirk-Cohen and Plants 2019) show a high level of stress among students during their university studies. According to these surveys, one third of students in higher education experience frequent periods of stress. This stress can be caused by a number of factors. Some of these may include how the university functions, exam periods, relationships with teachers, the pressure that parents put on academic achievement, competition with other students, financial problems, uncertainty about the future, and lack of self-confidence (Croog 1970; Schonpflug 1985; Zapf et al. 1999; Ross et al. 1999; Abdulghani 2008).

Certainly, explains Fleming (1981), the transition from adolescence to adulthood is a difficult time for several reasons. For example, leaving the parental home can be a distressing experience for some students, gradually building a professional identity is a long and uncertain process, and social isolation can lead to serious psychological problems. All studies highlight the harmful effects of chronic stress. Vulpe (2021) shows that the effects of stress are characterised by a decreased ability to make rational decisions, a tendency to make mistakes, a shortened attention span, difficulties concentrating, mental blocks and hypersensitivity to criticism. Situations of stress can heavily increase situations of students dropping out of university, feeling worried and insecure about their level of knowledge, a significant increase in cigarette and alcohol consumption, insomnia and loss of appetite (ibid.). Laurence et al. (2009) observe that women generally feel more stressed than men. Early studies (Rim 1986; McCrae and Costa 1986) highlighted how introverts are more susceptible to stress, anxiety and pessimism. More recent studies (Iamandescu 1993; Bunevicius et al. 2008; Hagger 2015) also note a strong correlation between personality types and stress. While stress is characterised by a tendency of impulsivity and aggression among extroverts, Iamandescu (1993) highlights that excessively repressing emotions can lead to depression in introverts. Stress among students thus has many consequences and their effects have a significant impact. According to Schneider et al. (2012), students suffering from stress tend to avoid new or complex tasks. They may also no longer be able to develop their professional skills. The demands and expectations from teachers and parents and even the competitive spirit among students all lead to high levels of stress, a lack of motivation to learn new things or a loss of interest and thirst for knowledge (Fornés-Vives et al. 2012).

Of course, stress is not a negative thing in itself. It is part of everyday life (Selye 1973) and can have a positive impact if it is managed properly (Bull 1991; Franke 2014). Matei (2006) points out that stress can be useful and necessary for improving the learning process of knowledge and know-how. Furthermore, it can make learning faster when at an acceptable level, but beyond that and in the long term, studies show that stress is detrimental to students' physical and mental health. It affects the immune system (Wyman et al. 2007; Caserta et al. 2008; Fagundes et al. 2013) and leads to an increased risk of disease (Bick et al. 2012).

## 2. Romanian Students Are Experiencing Stress

When it comes to stress, Romanian students are no exception, especially because of the particularly volatile situation faced by the nation. This instability is political, economic and psychological (inflation, skilled labour immigration, depreciation of the national currency,

deterioration of the labour market). Over the last 28 years, Romania has had 24 ministers of education—including four in 2012 alone—and so these frequent changes in education have led to inconsistency in government schemes and have required students to make considerable efforts to adjust to these changes (Antoci and Mafteuta 2017). Like their European counterparts, Romanian students are faced with a great deal of stress. A study conducted on a sample of 100 Romanian first-year medical students (Mihăilescu et al. 2011) shows that high levels of stress negatively impact learning and lead to depression and, ultimately, to the most vulnerable students dropping out of university. Uncertainty about the future is also a major cause of stress (Balgiu 2014). In another study conducted with French, Moldovan, and Romanian students, Habihirwe et al. (2018) report stress and anxiety rates of 39%, 47%, and 35.8%, respectively. According to a recent study (UNSR 2021[1]), 48% of Romanian students admit to being stressed on a regular basis. The proportion of students who report having mental health problems caused by events during their time at university has increased by 45% over the last decade (Evans 2013). According to Matei (2006), a quarter of students experience difficulty concentrating and 30% feel isolated. Students also claim to be stressed by the challenges of student life and the demands of the working world (Matei 2006). They have a constant fear of failure and doubt themselves, their academic skills and success in their careers after graduating. Attending classes, managing coursework and exam periods are generally considered very stressful, especially for students who also work. A study conducted by Săvescu (op cit) indicates that students who do not work are less stressed and perform better than those who do work. The same study indicates that in 2016, 45% of nonworking students passed all exams compared to only 26.5% of working students. For Romanian students, particularly stressful situations include health problems, work–life balance, fear of failure, and exam periods (ibid.).

The literature clearly shows the causal links between stress and student life. Our study aims to deepen our knowledge in this field in Romania. It successively examines stress factors, symptoms experienced by students, conditions for organising studies and ways to improve students' quality of life. This explorative study is further justified due to the specific context of the pandemic and lockdown, which had a hugely negative impact on students.

## 3. Methodological Framework

To conduct our study, we sampled students at the Dunărea de Jos University of Galați. We chose to carry out our study at this particular institute because the Dunărea de Jos University of Galați, which was founded in 1974, is a major university in Romania with more than 12,000 students. It is located in the city of Galați, which lies in eastern Romania, on the border with the Republic of Moldova. This university piqued our interest because it is considered an excellent university and has been awarded with the appellation "high degree of confidence." It has 14 faculties for all academic levels in a range of fields including technology, human resources, economics, the environment, and the arts. We first conducted a questionnaire with open-ended questions on the feeling of stress, its symptoms and its consequences, and more closed-ended socio-demographic questions (the questionnaire is attached in Appendix A). This questionnaire was tested with students on campus. We were able to better identify some important factors of stress that we added in our questions. Then, we distributed the online questionnaire via Google Forms to students enrolled in the five fields represented at the University of Galați. Our exploratory approach explains our choice to ask questions in a binary mode. Our objective was to obtain precise data allowing comparison between groups of respondents. In a second step, we intend to carry out a more in-depth and qualitative analysis through interviews and immersive observation. We decided to interview only first-year students, as this year is a new and stressful transition period for newcomers to the university. Answers were collected between March and April 2022, shortly after the pandemic and the return of face-to-face classes in Romania. In total, we collected 151 questionnaires, which we processed with Microsoft Excel and the content analysis software NVivo to identify units of meaning (words, terms, phrases). In our

sample (see Figure 1), 53.6% were female and 44.6% male (1.8% of respondents preferred not to answer this question), 70% were between 21 and 23 years of age, and 30% between 24 and 52 years of age. Because this was a cluster sample, we applied a weighting by subject area to reflect the different courses offered by the university. Respondents were enrolled in the arts (20% in physical education and sport), economics (23% in economics and business administration), technology (21% in computer science), human resources (10%), and the environment (6%). In terms of domicile, 65% of respondents were from urban areas and 35% from rural areas. It is important to note that this gap can be explained by the fact that most young people who complete secondary school in rural areas prefer not to continue their education beyond that point; 68% of our respondents were from the city of Galaţi, which is where the university is based. The second-most represented city was Brăila, located 20 km from Galați. Other students come from neighbouring regions, such as Buzău, Bacău, Vrancea, Tulcea and Vaslui. Some students come from more distant cities, such as Bucharest or Ploieşti. It is also worth noting that 49.7% of respondents were studying and working at the same time. In Romania, it is common for students to work alongside their studies because of the difficult economic and financial situation, as parents cannot afford study-related costs. Without these financial resources, a majority of students would not be able to support themselves. With regard to the nationality of the respondents, 134 were Romanian, 14 Moldovan, and 3 Ukrainian.

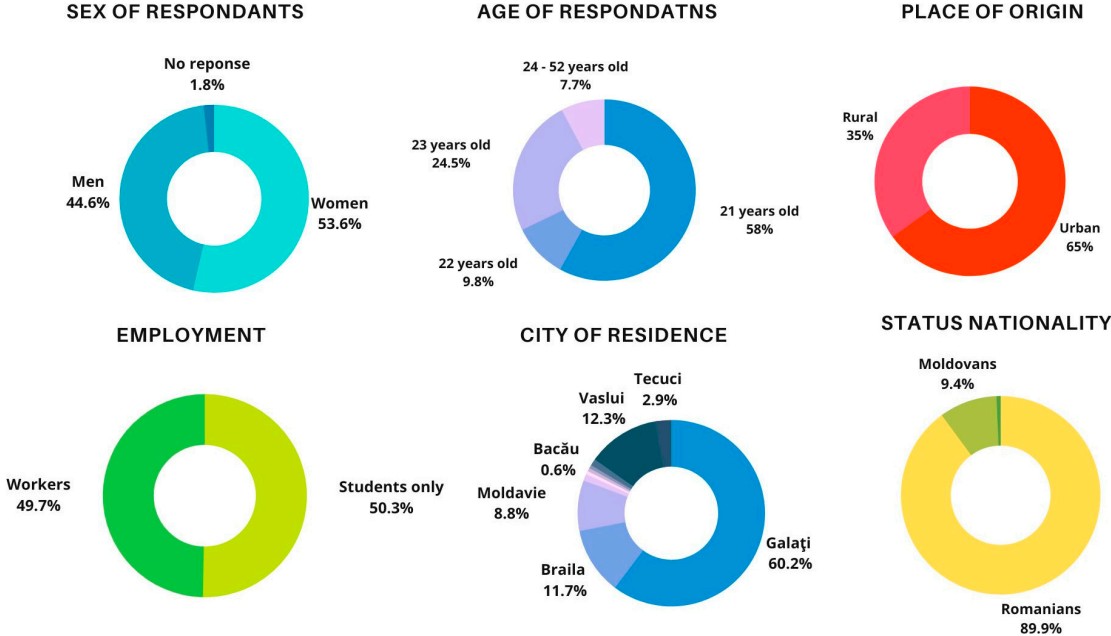

**Figure 1.** Socio-demographic data of students at the University of Galați. who responded to the survey.

## 4. Results, Analysis, and Discussion

Our survey allowed us to better identify the time-related, financial and social factors of stress and the resulting symptoms. It is very clear that the COVID-19 pandemic has had a detrimental effect on this already fragile young population.

### 4.1. Stress Factors and Main Symptoms

Archer and Lamnin (1985) claim that stress is caused by academic pressures (competing for high grades, often short deadlines, teachers' personalities, atmosphere of the study group, the need to successfully enter the world of work) and sometimes difficult circumstances in their personal lives (social and romantic relationships, problems with parents, financial problems or conflicts with friends). More recent studies have made the same observations. They all show a strong correlation between the level of academic achievement expected from the institute and the quality of student life. They show that

the causes of stress are related to the academic environment (exam periods, short revision periods, homework or group work, discipline structure), the teaching environment (assignments and obligations in the programme) and the students' personal and private lives. Murphy et al. (2009) observe that stress is caused by the culture of performance and academic achievement. Students are often concerned about relationships with the teaching staff, personal issues and the search for a professional identity.

Our study shows that 87% of students surveyed reported being stressed in their day-to-day lives (this is double the rate in the 2021 study by the Romanian National Students' Union (UNSR) previously cited). It confirms the surveys cited above. Our respondents mentioned many stress factors (Figure 2). Of these, time management—particularly relating to exam requirements—came out on top (66%). This pressure is further heightened for students who have to work alongside their studies (49%). This is followed by the country's economic situation (55%), uncertainty about the future (52%) and social and intergenerational pressures (28.5%).

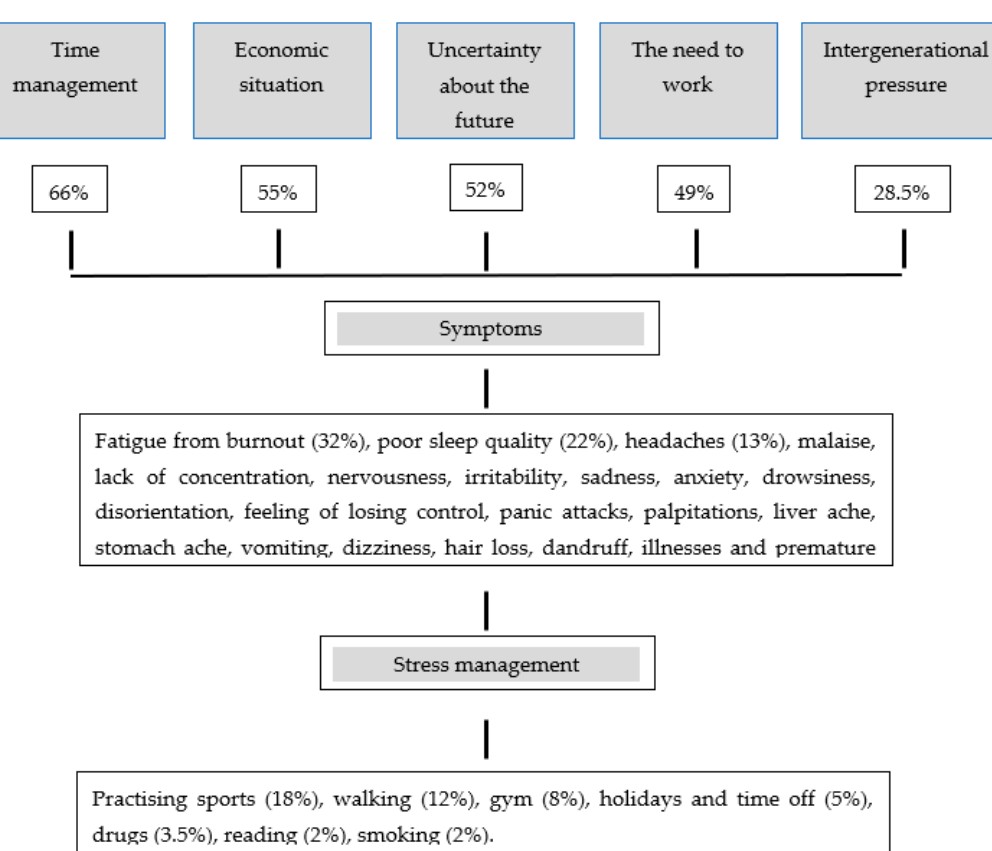

**Figure 2.** Stress factors, symptoms and ways to manage them.

In sum, 66% of the students who responded to our questionnaire stated that they often felt stressed because of the pressure of their busy schedules. While they find it difficult to organise their time between classes, seminars, projects, and writing their dissertation, many have to balance their responsibilities as a student with those in their private, professional, and family life. "The most important stress factor is the busy schedule: exams, dissertation, projects, work", say our interviewees. Many feel confused, they do not know what is most important to them, and are unable to prioritise. One student said, "Since starting my studies, the busy schedule has left me exhausted, my mental performance has decreased, I'm making a lot of mistakes, and my mental health is low." In addition to these time constraints, 49% of students have to also manage professional obligations as they work alongside their studies. One student stated, "It's difficult to concentrate on your studies when you have to work to support yourself". Furthermore, 55% feel stressed due to the economic and social situation in Romania. One respondent said, "It's a disastrous economic

situation because most Romanians have low standards of living and no money—their salary isn't enough". Another student claimed, "When I think about the costs after my studies and salaries for entry-level jobs, I feel like leaving Romania". Uncertainty about the future within a complicated socio-economic context is another major cause of stress for 52% of those surveyed. One student said, "My future career is stressing me out a lot. I don't feel ready at all, I don't know where to start." Another student added, "I'm stressed because I'm uncertain about my career, and I'm worried about not having enough income right after I graduate". Lastly, social and intergenerational pressure was a stress factor for 28.5% of our respondents. One student stated that "Our generations are very different and my parents have different requirements from what I want."

There are various symptoms that students experience as a result of stress. They are now well identified. Students suffering from stress report feeling unhappy, having stomach pains, difficulty relaxing and sleeping, mental health problems, and even depression (Dusselier et al. 2005; Hystad et al. 2009; Lungu 2021). Among those most commonly mentioned by respondents in our study were fatigue resulting from burnout (32%), poor sleep quality (22%) and headaches (13%). Other common symptoms included malaise, lack of concentration, nervousness, irritability, sadness, anxiety, drowsiness, disorientation, feeling of losing control, panic attacks, palpitations, liver pain, stomach ache, vomiting, dizziness, hair loss, dandruff, illness and premature greying. Interestingly, stress affected respondents' appetites differently: a number of respondents reported a loss of appetite (8%), while some claimed to have an increased appetite (2%).

Our results mirror those of Săvescu et al. (2017). To manage their stress, students participate in recreational activities including sports (18%), outdoor activities, such as walking (12%), or exercise at the gym (8%). They also take advantage of short breaks and time off (5%) when they are able. Some students (2%) reported "reading books to better manage their emotions". They also take care of their diet, talk to their family and friends, watch films and listen to podcasts. However, not all these stress management activities are beneficial. Some students admitted to taking up addictive habits, such as drugs (3.5%) and smoking (2%). However, all these "strategies" are not enough to reduce stress. Some students said that they have not yet "found the solution"; one said, "I can't really handle the stress". Another stated, "I'm living in the now and I'll deal with the consequences later".

### 4.2. Difficulty with Time Management

As previously mentioned, the students interviewed about stress factors first talk about managing their time, which is split between learning, revising and assessment periods. Historical studies already showed a strong correlation between social and institutional pressure and the levels of depression, and even suicidal tendencies, observed among students (Benfari et al. 1972; Fleming 1981; Archer and Lamnin 1985; Blumberg and Flaherty 1985; Coyne 1985). Two thirds of students in our study stated that they have difficulty organising their work, revision, and preparation time, regardless of whether this involves completing projects, writing dissertations, preparing for exams, or the exams each semester. They become stressed because they do not have enough time. They struggle to prioritise their deadlines. The consequences of their stress may be cognitive for some or felt bodily by others. One student talked about feeling disoriented and being unable to be organised and prioritise: "I don't know where to start". Another student reported sometimes being "unable to speak without stuttering". While most students see teachers as a source advice, encouragement, and putting them in touch with employers, one quarter sees teachers as an additional cause of stress. Teachers have many requirements that students must meet in order to pass exams. For these students, teachers "stress them out more than they help them". They want professors to be more lenient, especially in the first year (where students are discovering a whole new world) and in the final year, which is decisive for completing their course of studies. Students reported having more and more work to complete as they progress through their years of study, due to a hectic schedule. Students noted "a lack of education on managing their time" and "a fear of failure" that is further heightened by

the remarks and the attitude from some of the teaching staff. They believe that earlier and more thorough planning of tasks would help them better manage their stress.

### 4.3. The Need to Work While Studying

The economic and financial situation in Romania does not allow all students to focus solely on their studies: one student said, "I don't have enough income to not work". "The economic and social situation in Romania is seriously flawed", said another respondent. A majority therefore need a job that will provide them with the necessary income to support themselves. Furthermore, it should be noted that some students have families to support, which requires them to find ways to secure the financial resources they need. These findings explain why working students make up the majority in our sample. As a matter of fact, while 35% of the students surveyed stated that they did not work during their studies, almost 50% reported having a job while studying. It should be noted that the remaining 15% was made up of students who had a job but had to quit to concentrate on their studies (they do, however, work part time or occasionally). Among working students, a vast majority (90%) claimed to be very stressed and had difficulty completing their final year of studies (note that previous studies have already shown this high level of stress among working students—see Home 1997, for example).

### 4.4. Uncertainty about What the Future Holds

Future careers are another major cause of stress for more than half the students surveyed. The job market in Romania is quite complex: there are few jobs, and those available for young graduates are not paid enough that would allow them to earn a living. There are many young people who do not know exactly what they want from life and do not know what path to follow. They believe that what they learned in university is not enough to get a job or to be successful. It is worth mentioning that in Romania, there are very few orientation and career development modules, and many students consider going abroad to earn a better living. Some Ukrainian students we met also talked about other causes of stress linked to the global situation: "I'm very worried about the situation in my country of Ukraine; I can't concentrate at university", said one. This point, however, remains a minority, as the percentage of students from Ukraine is very low in our sample.

### 4.5. Intergenerational Social and Cultural Pressures

To understand the stress that students feel, we must also consider the relationship they have with their parents and other family members. While a majority of students consider parental influence to be positive, nearly 30% feel the opposite. Although the former say they feel supported and helped by their parents, the latter talk about generational conflicts: "It is difficult to explain current events to older people", "Parents are stuck in the past and cannot understand current events", "The conflict between generations is caused by too many different points of view and a lack of open-mindedness". This generational gap (heightened by the exponential development of digital tools) is exacerbated in Romania by the still very present history of the communist era. While today's students are enjoying a period of economic freedom that offers them many opportunities, it is a different story for the older generations. Other problems in the relationship between parents and students arise from a lack of financial resources. The fact that parents are unable to support their children financially pushes the latter to work against their will. It should be noted that although intergenerational relationships appear complicated, relationships between students are very positive. While many studies have shown that medical, pharmacy, and dental students are especially likely to experience stress due to the intense workload in the first year of university (Beck and Srivastava 1991; Coburn and Jovaisas 1975; Davis 1989; Helmers et al. 1997; Vitaliano et al. 1989), this is not the case for our Romanian students. In fact, 80% of our respondents feel that other students do not cause stress.

### 4.6. The Traumatic Impact of the COVID-19 Pandemic

Our questionnaire understandably referred to the pandemic. The first studies show that the health crisis has had a significant impact on the lives of the population, particularly young people. According to a recent survey carried out for the journal *L'Etudiant*[2], the first confinement has greatly affected students, causing a growing sense of unease among the majority of them. A Fages-Ipsos study (2020) concludes that 76% of young people and 83% of students say they have been affected psychologically, emotionally or physically. The level of distress, the study notes, is worrying because more than a quarter of these young people (27%) and more than a third of students (31%) say they have had suicidal thoughts since the beginning of the crisis (a proportion that has risen sharply in 9 months). A study published in 2021 (Macalli et al. 2021) shows similar results. During the period of confinement, students were more affected by mental health problems than non-students of the same age (37% and 20% respectively). On this subject, opinions of our respondents are unanimous: 90% of students had a very bad experience of the lockdown. There was admittedly a small number of students who feel that the pandemic made their studies easier in that they no longer had to waste time commuting to university. It is a well-known fact that many students have to travel long distances to get to university. During lockdown, students felt they had more time. Some scaled-down online courses allowed them to stay better organised and better manage their schedule. Some classes could also be recorded and watched later. Another advantage was the ability to speak online rather than in class, as some students have difficulty speaking in public. As such, while some students found value in the pandemic, a large majority expressed a strongly opposing view. It is apparent that the students we interviewed were deeply affected by the pandemic. Many consider this sudden change in their student life to be a shock, a trauma. Their responses illustrate this: "Our student years were taken away from us", "We've lost all the good times we could have had at university", "It was a terrible time of feeling insecure and worried", "I felt like my life was pointless and I lost my motivation for university", "It was the hardest time of my life", "It really tested the limits of my mental health". All our respondents acknowledged that they suffered from a lack of social interaction. Especially for those who are active and sociable, the pandemic has been very stressful, some even say "traumatic" (nervous tremors, tearfulness, memory loss). Some students (27%) reported not learning anything, which affected their performance and mental health. The lack of interaction with their teachers was often difficult. Admittedly, the transition to online learning was not always well managed by institutes and the teaching staff. The different ways of teaching and different programmes have created major inequalities. While the computer science students appreciated being able to continue their courses online, students taking other courses felt deeply unsettled by practical work and tutorials being cancelled.

### 4.7. A University Policy to Be Reviewed

The Romanian legal and institutional framework does not include any specific provisions to help students manage their stress or time. It also does not specify the amount of intellectual effort expected from students. The few items of legislation that address the amount of time students should spend on educational activities are not very specific. The January 2011 Law of National Education simply states that "a student's individual work may not be less than that which corresponds to 60 transferable academic credits each year". It also states that the student cannot "participate in learning for more than 8 hours per day" (this includes classes, lab time and seminars). The law also provides for a limit on the amount of time needed to complete homework, which must not exceed two hours per day for all subjects. However, this limit is relative and impossible to quantify in reality. On the other hand, students can receive financial support for transport and for participating in various cultural events. In some situations, Romanian students can receive merit-based scholarships, study grants or excellence scholarships. At the Dunărea de Jos University of Galați, the problem of time management remains strongly linked to exam periods. There are two exam periods during the academic year, each lasting three weeks (a winter session

in late January and a summer session in June). These exam periods can lead to fatigue, stress and psychological issues. As it is aware of these difficulties, the University of Galați offers short periods at the beginning of the year that allow students to learn about the university and how it operates. The university also offers cultural, artistic, educational and sports activities to help students relax and better manage their stress and fatigue. If better management of studies and exam periods by the university proves necessary, other measures can be taken. In this regard, our respondents wish they had more psychological support. Most want psychological, professional and educational advice on ways to best manage their time. In our questionnaires, the students highlighted the sometimes-difficult relationship with their teachers, and in some circumstances, a lack of empathy on their part. They want better communication. They are calling for a more effective relationship of trust with their teachers, so that they can be better heard, express their problems and find ways to solve them. They also complain that meetings and discussions with former students are not systematically organised. Equally, they are keen for meetings and exchanges with employers to take place in order to understand their expectations and the skills they need. One of our respondents shared how the students feel: "If I could, I would organise leisure activities within the university once a month, where students would be able to socialise, play some games, meet teachers and employers and talk to each other about their problems".

## 5. Conclusions

If our study confirms what other surveys show in different countries around the world, it also highlights the specificities of the Romanian context. The social and economic situation in Romania makes the stress level among Romanian students higher than in other European countries. To combat this, it is very important for academic and institutional players to understand and clearly identify the causes of stress so that they can offer adapted and long-term solutions. As our study points out, there are many factors of stress. They are social: our survey confirms the key role of parents and friends, who sometimes cause pressure and anxiety. They are economic: the difficulty of studying while also working is often insurmountable, which can lead to students quitting their studies and suffering from depression. They are also, and above all, structural: the students' responses show that institutes need to better address the management of university term time between preparation, learning, revision and assessment. In Romania, as in other countries, students need to be given assurance that they will have job opportunities and be well paid. Their living conditions must be better taken into account and their fear of uncertainty in their future needs to be better valued, so that the stress experienced does not become an insurmountable obstacle for the future of younger generations.

**Author Contributions:** Conceptualization, D.L. and D.B.-M.; methodology, D.B.-M. and D.L.; software, D.L.; validation, D.B.-M. and D.L.; formal analysis, D.B.-M. and D.L.; investigation, D.L. All authors have read and agreed to the published version of the manuscript.

**Funding:** This work was supported by the URePSSS laboratory and the SHERPAS team and Funding for Open Access Charge by the University of Lille.

**Institutional Review Board Statement:** Not applicable.

**Informed Consent Statement:** Not applicable.

**Data Availability Statement:** Not applicable.

**Conflicts of Interest:** The authors declare no conflict of interest.

## Appendix A. Model Questionnaire for Galati Students

**Survey at the University of Galati on student stress.**
**Interview guide**

1. Do you often feel stressed in your daily life?
   Yes ☐    No ☐

2. If yes, what are the factors that cause your stress?
......................................................................................................................................................................................
......................................................................................................................................................................................

3. Can you establish a hierarchy between all these factors?
   1. ........................................................................
   2. ........................................................................
   3. ........................................................................
   4. ........................................................................
   5. ........................................................................

4. Can you describe the symptoms of your stress?
......................................................................................................................................................................................
......................................................................................................................................................................................

5. Are you able to manage your personal and academic time?
   Yes ☐    No ☐

If not, can you explain why?
......................................................................................................................................................................................
......................................................................................................................................................................................
......................................................................................................................................................................................

6. Do you work or have you worked during your studies?
   Yes ☐    No ☐
If yes, would you say that your professional activity is a problem for your studies? Why?
......................................................................................................................................................................................
......................................................................................................................................................................................
......................................................................................................................................................................................

7. Are you worried about your professional future?
   Yes ☐    No ☐
If yes, can you explain why?
......................................................................................................................................................................................
......................................................................................................................................................................................
......................................................................................................................................................................................

8. Are you under pressure from friends, parents, other students?
   Yes ☐    No ☐
If yes, what kind of pressure is it?
......................................................................................................................................................................................
......................................................................................................................................................................................
......................................................................................................................................................................................

9. Would you say that the health crisis and the period of confinement have changed your stress level? Can you explain?
......................................................................................................................................................................................
......................................................................................................................................................................................
......................................................................................................................................................................................

10. How do you try to manage your stress?
......................................................................................................................................................................................
......................................................................................................................................................................................

11. Do you feel that this is enough to manage your stress?
   Yes ☐    No ☐
If not, why?
......................................................................................................................................................................................
......................................................................................................................................................................................

12. Do you feel sufficiently supported by your university and teachers?
   Yes ☐    No ☐
If not, why?
......................................................................................................................................................................................
......................................................................................................................................................................................

Free comment: Do you have anything else to say about your stress?
......................................................................................................................................................................................
......................................................................................................................................................................................
......................................................................................................................................................................................
......................................................................................................................................

**We want to know you better (we guarantee anonymity)**

- How old are you? ...................
- In which courses are you registered? ................
- What town or village are you from? ....................
- What is your nationality? .....................

Thank you for your participation.

## Notes

1    https://unsr.ro/2021/02/22/stresul-in-viata-studenteasca/ (accessed on 18 September 2022).

2    Survey for the website l'Étudiant conducted by Petitdemange A. in November 2020. Health crisis: the precariousness of students is increasing https://www.letudiant.fr/lifestyle/aides-financieres/crise-sanitaire-la-precarite-des-etudiants-augmente.html (accessed on 20 Septembre 2022).

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
