# Peer review of "Stress among Students and Difficulty with Time Management: A Study at the University of Galați in Romania"

_socsci, doi:10.3390/socsci11120538_

Round 1

Reviewer 1 Report

The paper is interesting but should be improved significantly:

- Social or sociological dimension is not clear in the abstract, and should be improved, especially in the introduction

- According to me, the abstract should be structured in the "Intro, Method, Results, Discussion, Conclusion" mode. This would be good for this kind of paper

- Some statements are not supported by empirical or bibliographical references. For example on page 2: “When it comes to stress, Romanian students are no exception, especially because of the particularly volatile situation faced by the nation". This statement, for example, is not supported by any reference, and seems almost like a judgement of merit.

- Fig 1 is in French

- The methodological part should say something more about questionnaires

- The discussion should contain more references to the literature review

- There are some bugs on bibliography, some references are not in alphabetic order

Author Response

Dear colleague,

Thank you very much for your serious and professional review of our text. We have carefully read your criticism and made the many changes and modifications listed below. We hope that these changes have significantly improved our text and thank you again for your patience.

The authors of the text

Summary

- We have better contextualised the sociological dimension of our study.

- We have modified our summary to better show our methodology and the main results, discussions and conclusions of our study.

Introduction

- We have better defined the stress and the particular situation in Romania.

- We have added many authors and references to better argue our text

Methodology

- We have made the content of the questionnaire more explicit and attached it as an annex

Results and discussion

- We have better referenced (authors and statistics) some parts of the text especially on the economic, political and social context of Romania

- We have translated Figure 1 into English.

- We have moved some paragraphs from the introduction to the discussion part and added other documentary references to better contextualise our results in relation to other surveys in other countries.

- Wherever necessary, and at every point highlighted by the reviewers, we have added references to support our statements.

Bibliography

We have corrected the bibliography, added new references, respected the alphabetical order and harmonised the whole according to the APA style.

Reviewer 2 Report

Dear author(s)

Thank you for giving me the opportunity to read your work. 

I think you contribute a valuable perspective from Romania, and the topic of stress in relation to (political and educational) environments seems relevant. Unfortunately, I am not convinced that the paper is ready to be published. I have stated concrete points below. Major challenges are, that the theoretical part needs to be structured more clearly and lacks several concepts and definitions. The operationalization and scale are not transparent. The method is missing, the reader can not follow the analysis and results. The discussion is intertwined with the results, I suggest splitting those sections. The conclusion seems normative.

I hope my review can help you to improve our manuscript and encourage you to continue working on the contribution! 

Concretely, I suggest:

  • Native speaker proofread

  • “Regardless of gender or age, more and more people are suffering from stress, for which there are many causes and effects.” Please add the increase and reference.

  • “Golu (1991) defines stress as a state of discomfort and tension caused by external factors.” Okay, maybe he does this, but that's not the only way to see stress, internal factors also impact stress (resilience).

  • “This author believes that stress has a negative impact, which leads to frustration and increased difficulty in addressing issues.”. This is unconventional: concluding/opinion in the theoretical part of the author and I believe there is research on the negative effects.

  • “Some of these may include how the university functions, exam periods, the relationship with teachers, the pressure that parents put on academic achievement, competition with other students, financial problems, uncertainty about the future and lack of self-confidence.” It would lift the paper to add references with studies showing such effects, maybe include eco-anxiety to the list, one could also call out the bigger phenomenon like neoliberal-capitalist conditions of life.

  • Please state how stress can be managed properly and at an acceptable level, both sound vague.

  • “But beyond that and in the long term, studies show that stress is detrimental to students’ physical and mental health.” Please add references and be precise about what the long run means. Isn't acceptable levels of stress and correctly handled stress healthy enough also in the long run? 

  • Is 1980 chosen for a reason?

  • What are serious psychological problems? 

  • How is stress in the first semester already chronic? Please be precise. 

  • Can you elaborate more on the situation in Romania? What do the many shifts in political positions mean, and what are the symptoms, is there a larger picture? 

  • I suggest restructuring and picturing more clearly what is typical about Romanian students and what is special. The section is hard to follow. 

  • Please make your items transparent/questionnaire, did you use an established scale, and if not, why not? 

  • Please explain what you mean by testing, factor analysis? What are the results? 

  • “Our survey, which includes Romanian students in GalaÈ›i” I think the reader got that by now, I suggest deleting this repetition. 

  • How do you operationalize stress? 

  • I wonder how far the items are suggestive, do you suggest “time management”?

  • How do you operationalize “often”

  • I actually don't understand the analysis, which methods did you use, some seem descriptive statistics, how did you analyze the open-ended questions? Sometimes I see percentages, but I miss the method.

  • “While they find it difficult to organize their time between classes, seminars, projects, and writing their dissertation, many have to balance their responsibilities as a student with those in their private, professional, and family life.” Can you give examples from the data and organize them alongside concepts on a more abstract level? 

  • I value the data examples given, very interesting

  • I am a bit confused when you switch to discussing the results, I suggest making clear sections. I don't follow some concepts, like “traumatic”, did you diagnose that?

  • The conclusion formulates strong imperatives, which I don't always see supported by the analysis but normative, moreover, it seems to merely summarize the introduction, if the data is identical with the entire state of research (which can happen) this should be framed as such. 

Author Response

Response reviewer 2

Dear colleague,

Thank you very much for your serious and professional review of our text. We have read your criticism carefully and made the many changes and modifications listed below. We hope that these changes have significantly improved our text and thank you again for your patience.

The authors of the text

Methodology

- We have made the content of the questionnaire more explicit (open questions, closed questions, typology of answers).

- We have attached the questionnaire as an annex for greater clarity.

- We have clarified the stages of our methodological approach (selection of the sample, testing, results and analysis).

- We have clarified how, by testing the questionnaire, we were able to add some questions on important stress factors.

Results and discussion

- We have better defined stress by adding the principle of resilience.

- We have moved paragraphs from the introduction to the discussion section and added further literature and authors references to better contextualize our results in relation to other surveys in other countries.

- Wherever necessary, and at each point highlighted by the reviewer, we have added references to support our observations (statistics on stress in general, acceptable levels of stress, etc.).

- We have added references to studies showing the forms of eco-anxiety and stress related to the more neoliberal and capitalist living conditions that have transformed Romania.

- We have clarified the nature of the serious psychological problems observed following prolonged periods of stress

- We have better explained why stress in the first year is particularly important for newly arrived students

- We have better referenced (authors and statistics) some parts of the text especially on the economic, political and social context of Romania

- We tried to better specify the context and the nature of the stress experienced by Romanian students (economic context, uncertainty about the future, parental pressure...).

- We have added examples and verbatims to illustrate the difficulties (notably time management, social pressures) lived by the students.

Conclusion

- We have improved the conclusion to better single out the situation of Romanian students and to better highlight the specific context of their studies.

Round 2

Reviewer 1 Report

Ok for me

Author Response

Thank you very much for your help in the revision of our manuscript.

Best regards

The authors

Reviewer 2 Report

Dear authors

Thank you for the revised version of your manuscript. I see that you have put a lot of effort in the revision and the manuscript indeed did improve. However, for a publication in a scientific journal, I need to stand on some more improvements. Most pressing is the lack of a method. Another problem is the suggestive conclusion formulating implications for e.g. the government. 

I believe that your paper is a valuable contribution if presented with a method and without normative implications. 

For the future, I warmly suggest using established scales for data collection (for the quantitative parts, or going completely explorative). To me, it seems very unusual (and problematic) with the binary answer options instead of a Likert scale: One can not manage academic time or not, it's progressive). However, I think the Romanian data would contribute to the state of research, again, if analyzed methodically controlled. 

Please see my concrete critique and examples below. 

  • “Today, negative environmental factors such as climate change, war, and health crises have consequences for the morale of populations.” Maybe delete morale? It has consequences on populations, and I would not relate stress to morale.

  • Still, the method for analysis is missing, if this is descriptive statistics, you need to state that. And it seems that you used some sort of content analysis for the open-ended questions. The data examples are convincing and interesting, nevertheless, in an academic/scientific contribution, a method and analysis are mandatory. On page 6 I see categories that make sense, but you need to explain how you (methodically controlled) got to these results (content analysis/thematic analysis?)

  • I still wonder whether the survey is building on a scale and if not, why so? If it is an explorative study, that is fine and legit, but necessary to state and argue explicitly. The “yes” and “no” binary instead of Likert scales seem special, please argue why to choose this design or how it happened (it can happen that we collect data, that are not perfect, and we can still use such data, but we need to explain how come and be transparent and reflexive about it).

  • “Old studies” I would use “historic” instead of old. 

  • “They become stressed because they do not have enough time. They struggle to prioritise their deadlines. One student talked about feeling disoriented and being unable to be organized and prioritise: “I don't know where to start”. Another student reported sometimes being “unable to speak without stuttering”.” I suggest all results have an explicit structure by clustering and sorting, in this example some effects are cognitive interpretations some are bodily effects, that look (to me) like subcategories.

  • “The social and economic situations in Romania make the stress level among Romanian students higher than in other European countries.” Seems to make it even stronger. if you want to directly compare, you need to measure the degree, and for that, a scale would be necessary. Your questionnaire can not control/ measure the degree with the binary answers.

  • “Their living conditions must be better taken into account and their fear of uncertainty in their future needs to be better valued. Governments must become aware of the urgency of the situation and act accordingly so that the stress experienced by students does not become an insurmountable obstacle for the future of younger generations” Again: I strongly recommend leaving normative suggestions out of any scientific paper. It's fine concluding that they are under pressure and why so. What governments have to do, is not to be placed here. 

Author Response

Dear colleague,

We sincerely thank for your reading and comments. We have taken these criticisms into account and have made the following changes:

- We have improved our methodology. We have clarified that the Nvivo software allowed us to identify the units of meaning.

- We have explained our choice of binary questions in this exploratory study. We also explained that this study would be the first step in a future more qualitative and immersive study.

- We removed some words or expressions (the 'moral' state of the students). We replaced some words and expressions ("old studies" by "historic studies").

- We have better presented the verbatims and distinguished between cognitive and physical consequences.

- We have removed in the conclusion sentences that were too "prescriptive" in relation to government policy.

We hope that we have met the reviewer's requests and thank him again for his patience and pertinent remarks.

The authors

Round 3

Reviewer 2 Report

Dear authors

Thank you for clarifying and improving your manuscript. I do look forward to seeing it published after this process. 

Kind Regards